

# Looking beyond general metrics for model comparison - lessons from an international model intercomparison study

Tanja de Boer-Euser[1], Laurène Bouaziz[2], Jan De Niel[3], Claudia Brauer[4], Benjamin Dewals[5], Gilles Drogue[6], Fabrizio Fenicia[7], Benjamin Grelier[6], Jiri Nossent[8,9], Fernando Pereira[8], Hubert Savenije[1], Guillaume Thirel[10], and Patrick Willems[3,9]

[1]Water Resources Section, Faculty of Civil Engineering and Geosciences, Delft University of Technology, P.O. Box 5048, NL-2600 GA Delft, The Netherlands
[2]Deltares, Hydrology, Delft, Netherlands
[3]Hydraulics division, Department of Civil Engineering, KU Leuven, Kasteelpark Arenberg 40, BE-3001 Leuven, Belgium
[4]Hydrology and Quantitative Water Management Group, Wageningen University, The Netherlands
[5]University of Liège, Liège, Belgium
[6]Laboratoire LOTERR, Université de Lorraine, Metz, France
[7]Eawag, Dübendorf, Switzerland
[8]Flanders Hydraulics Research, Antwerp, Belgium
[9]Vrije Universiteit Brussel (VUB), Department of Hydrology and Hydraulic Engineering, Brussel, Belgium
[10]Irstea, Hydrosystems and Bioprocesses Research Unit (HBAN), 1, rue Pierre-Gilles de Gennes, CS 10030, 92761 Antony Cedex, France

*Correspondence to:* Tanja de Boer-Euser (t.euser@tudelft.nl)

**Abstract.** International collaboration between research institutes and universities is a promising way to reach consensus on hydrological model development. Although comparative studies are very valuable for international cooperation, they do often not lead to very clear new insights regarding the relevance of the modelled processes. We hypothesise that this is partly caused by model complexity and the comparison methods used, which focus too much on a good overall performance instead of fo-

5  cusing on specific events. In this study, we use an approach that focuses on the evaluation of specific events and characteristics. Eight international research groups calibrated their hourly model on the Ourthe catchment in Belgium and carried out a validation in time for the Ourthe catchment and a validation in space for nested and neighbouring catchments. The same protocol was followed for each model and an ensemble of best performing parameter sets was selected. Although the models showed similar performances based on general metrics (i.e. Nash-Sutcliffe Efficiency), clear differences could be observed for specific

10  events. The results illustrate the relevance of including a very quick flow reservoir preceding the root zone storage to model peaks during low flows and including a slow reservoir in parallel with the fast reservoir to model the recession for the Ourthe catchment. This intercomparison enhanced the understanding of the hydrological functioning of the catchment and, above all, helped to evaluate each model against a set of alternative models.

## 1 Introduction

15  The world we are living in is under constant change at different temporal and spatial scales. Growing populations and urbanization lead to changing land use, whereas climatic change may lead to more extreme events in the future (IPCC et al.,



2014). Considering that hydrological predictions are used by many societal sectors dealing with water allocation, ecosystems, droughts and floods, and the increasing economic value of assets in flood prone areas (e.g., UNISDR, 2011; Beckers et al., 2013), it is important to decrease predictive uncertainty now and in future conditions (e.g., Ehret et al., 2014; McMillan et al., 2016).

To make good predictions under different conditions, understanding of the hydrological functioning of a catchment is essential (Kirchner, 2006). The outflow from a catchment is a combination of different runoff processes, occurring in different parts of the catchment and at different moments throughout the year (e.g., Berghuijs et al., 2014; Nippgen et al., 2015; Penna et al., 2015). Threshold behaviour (e.g., Spence, 2010; McMillan, 2012) and heterogeneity of moisture states (e.g., Detty and McGuire, 2010; Rinderer et al., 2014) create complex systems from which it is difficult to filter the relevant timescales and

processes. Therefore, trying to understand and filter these relevant processes is an ongoing aim of modelling and experimental studies (e.g., McMillan et al., 2011; Kumar et al., 2013; Hrachowitz et al., 2016; Nijzink et al., 2016).

Hydrological studies at different scales and under different climates have shown a large variety of hypotheses on hydrological functioning (e.g., McDonnell, 2013; Zehe et al., 2013; Fenicia et al., 2013; Clark et al., 2016; Seibert et al., 2016). This variety of hypotheses and modeller's experiences has led to a wide variety of models; models which are tailored to specific

areas, available input data and required output data (e.g., Gassman et al., 2007; Samaniego et al., 2010; Clark et al., 2015; Maxwell et al., 2015). Overall, models vary in process representation (conceptual vs. physically based), in the degree of spatial distribution (lumped, semi-distributed and fully distributed) and in the actual runoff process being modelled (e.g., Fenicia et al., 2016). The disadvantage of this abundance of models is that new insights and developments are very scattered and difficult to combine (e.g., Weiler and Beven, 2015). However, a large advantage of having all these different models is their possible

use as multiple working hypotheses (Clark et al., 2011) in a model comparison study to investigate which processes, process representation and spatial distribution are suitable for a set of catchments.

Comparative studies are common in hydrological science and each study has its own twist. While some studies may focus on simulations in a large variety of catchments with widely different characteristics (e.g., Gupta et al., 2014; Duan et al., 2006; Gudmundsson et al., 2012), others focus on a variety of model structures in a limited number of catchments (e.g., Holländer

et al., 2009, 2013; Nicolle et al., 2014; Koch et al., 2016; Breuer et al., 2009; Vansteenkiste et al., 2014). Many of them rely on international collaboration between several institutes and universities to tackle important open hydrological research questions. Important findings from large sample studies are the possibility to rigorously test alternative model hypotheses and to derive the range of applicability of model structures for certain catchments (e.g., Gupta et al., 2014; Thirel et al., 2015b, a). A lesson learned from comparative hydrology in a small number of catchments is the importance of soft data (modeller's

system understanding) as well as hard data (data and model), as among others described by Winsemius et al. (2009) and Holländer et al. (2013). Ceola et al. (2015) concluded that deriving the causes of performance differences between various model structures is not trivial, mainly due to the considerable differences in model structures which disturbs the identification of model features that increase model performance. Nevertheless, comparative experiments with different model structures should be encouraged to maintain the dialogue between different research groups and agree on adequate modelling concepts

(Weiler and Beven, 2015).





During the last decade, model comparison studies have become much easier to carry out due to the large amount of freely available data and the increasing options for sharing data, tools and models. However, a solid model comparison study requires both a clear protocol, and a fair comparison method for the model results (Ceola et al., 2015). Protocols can, among others, contain information regarding pre-processing of data, calibration techniques or guidelines for transferring parameter sets. Very strict protocols do not always line up with the experience of the modeller and the different requirements for each model. Therefore, protocols should be clear, but can never be all-embracing. On the other hand, assessing the performance of the different model realisations should be identical. Standard performance measures (i.e., Nash-Sutcliffe Efficiency, Root Mean Squared Error, Mean Absolute Error) give a general overview, but are unable to point out small differences between model realisations (e.g., Schaefli and Gupta, 2007; Euser et al., 2015). The small differences can possibly be visualised by focussing on specific events and by using more specific performance indicators and data (e.g., Rakovec et al., 2016).

Thus, model comparison studies can be a powerful tool to maintain the scientific dialogue and may contribute to increasing catchment understanding. In this study, different universities and institutes working in and studying the transboundary Meuse basin, in Western Europe, applied their rainfall-runoff model to a set of subcatchments of the Meuse basin using the same meteorological forcing. Modelled discharges and internal fluxes were analysed to gain insight in the behaviour of a set of hydrological models. Our objectives are as follows: (i) propose a model intercomparison and evaluation protocol focused on the assessment of specific events instead of general metrics, and discuss the challenges associated to a general and objective approach to model evaluation, (ii) apply the intercomparison protocol to various hydrological models proposed by various international institutions, and (iii) relate differences in the simulated hydrographs to model components, and to their associated processes representations

## 2 Study areas and data

This study focusses on three subcatchments of the Meuse basin in the Belgian Ardennes: Ourthe, Lesse and Semois catchment and on the two main subcatchments of the Ourthe: Ourthe Orientale (eastern side) and Ourthe Occidentale (western side) catchments (Figure 1 and Table 1).

The Ourthe catchment at Tabreux was selected for calibration because of the limited influence of artificial reservoirs and its meso-scale which enables to focus mainly on hydrology instead of hydraulics. One large reservoir is located in the Ourthe catchment at the confluence of the Ourthe Orientale and Ourthe Occidentale; a short study showed that the influence on the downstream discharge is relatively small (see Section 10 of the Supplement for more explanation). The Ourthe is a typical rain-fed river with a fast response to rainfall due to shallow soils and steep slopes (Driessen et al., 2010) and has a strong seasonal behaviour (Euser et al., 2015).

Many studies have already been carried out in the Ourthe catchment (e.g., Driessen et al., 2010; Rakovec et al., 2012; Euser et al., 2015) because of its significant contribution of flow volumes in the Meuse during floods (de Wit et al., 2007). The catchment of the Ourthe at Tabreux has a total area of 1607 $\mathrm{km^2}$ with an elevation ranging between 107 and 663 $\mathrm{meters}$. Mean annual precipitation and evaporation are 1000 $\mathrm{mm\,y^{-1}}$ and 730 $\mathrm{mm\,y^{-1}}$ respectively. The main land use is agriculture (28%



crops and 28% pasture), followed by forestry (46%) and only 6% of the catchment has an urban cover (CORINE Land use map, European Environment Agency, 2006).

The neighbouring Lesse and Semois catchments and the nested Ourthe Occidentale and Ourthe Orientale catchments were selected for validation. The Lesse and the Semois catchments are about 25% smaller than the Ourthe catchment, and their forest cover is slightly higher than in the Ourthe catchment (Table 1). Annual mean precipitation is similar in the Lesse catchment while it is 25% higher in the Semois catchment. The upstream parts of the Semois catchment and the nested catchments within the Ourthe (Occidentale and Orientale) are relatively flat, while the Lesse catchment and downstream parts of the Ourthe and Semois catchments have steeper slopes.

Data preparation involved interpolation of hourly precipitation station data based on Thiessen polygons. The station data is collected and made available for this study by the Service Public de Wallonie[1]. Daily minimum and maximum temperatures from the freely available gridded E-OBS dataset (0.25° x 0.25° resolution; Haylock et al., 2008) were disaggregated to hourly values using radiation data at Maastricht (Royal Netherlands Meteorological Institute[2]) and a sine function. Daily potential evaporation was derived with the Hargreaves formula (Hargreaves and Samani, 1985) and disaggregated to hourly values using a sine function during the day and no evaporation at night. Precipitation and temperature data were available for the period from 1 January 2000 to 31 December 2010 (local time). The data (distributed, semi-distributed or lumped) was made available to the researchers through an FTP server. Figures of hourly observed discharge, precipitation, potential evaporation and temperature for each catchment can be found in Section 1 of the supplement.

## 3 Methods

This comparison study roughly consists of three elements: the modelling protocol followed by each participant, the models used by each participant and the tools used for comparing the individual model results.

### 3.1 Modelling protocol

Eight international research groups participated in this model comparison study using one or several rainfall-runoff models. A total of eleven models were used in this study, consisting of seven independent models and four models from the SUPERFLEX framework (Fenicia et al., 2011). A modelling protocol was prescribed to enable a comparison of the results. The protocol for the modelling involved a split-sample calibration and validation for pre-defined periods using a common dataset for the Ourthe catchment (Klemeš, 1986). The validation consisted of a blind validation in time (same catchment, but a different period) for the Ourthe catchment and a blind validation in space (same period, but different catchments) for the nested Ourthe Orientale and Ourthe Occidentale catchments and for the neighbouring Lesse and Semois catchments. Blind validation implies that only time series of forcing data (and no discharge observations) were given to the participants.

---

[1]Service Publique de Wallonie, Direction générale opérationnelle de la Mobilité et des Voies hydrauliques, Département des Etudes et de l'Appui à la Gestion, Direction de la Gestion hydrologique intégrée, Boulevard du Nord 8 - 5000 Namur

[2]http://www.knmi.nl/nederland-nu/klimatologie/uurgegevens, visited 14-12-2012





Calibration was carried out for the Ourthe at Tabreux for the period: 1 January 2004 to 31 December 2007, using 2003 as a spin-up year. Nash-Sutcliffe Efficiency (NSE; Nash and Sutcliffe, 1970) and NSE of the log of the flows (NSElog) were used as objective functions for calibration. NSE was chosen as a performance indicator for calibration because it is a common measure in hydrology to assess model performance with regard to high flows. NSElog was chosen as a second indicator to take low flow performance into account as well. Participants were free to use a calibration method of their choice to estimate parameter values. Although it makes the comparison less straightforward, a free calibration method accounts for the experience a modeller has with a specific model. Using the Pareto front, the best 20 realisations were selected for each model to account for a range in model realisations.

Validation in time was carried out for the Ourthe at Tabreux for the period from 1 January 2001 to 31 December 2003, using 2000 as a spin-up year. This period was selected for validation as it includes some relevant hydrological events such as the drought in the summer of 2003 and high flows during the winters. The validation period has relatively high flows compared to the calibration period. An additional validation in time was carried out for the period 2008-2010 for the Ourthe at Tabreux, using the calibration period as a spin-up. For the latter period, participants only received forcing data.

Validation in space was carried out for two nested catchments of the Ourthe: Ourthe Orientale (at Mabompré) and Ourthe Occidentale (at Ortho) for the period from 1 January 2001 to 31 December 2010, using 2000 as a spin-up year. Additionally, the derived parameter sets for the Ourthe at Tabreux were used to model the neighbouring Lesse (at Gendron) and Semois (at Membre) catchments for the same period. Only the forcing data was provided to the participants for this validation in space.

## 3.2 Descriptions of models

Each modelling group provided results as described above. A variety of models was used, including lumped, semi-distributed and fully distributed models. All models are conceptual, but their degree of complexity varies and they are used by institutes or universities working in the Meuse basin. Figure 2 depicts the main fluxes and storages of the applied models. Table 2 shows for each model the used forcing, the calibration method, whether parameters were regionalised and the institute which applied the model. Below a short description is given for each model: the term 'root zone storage' is used for the reservoir from which transpiration is modelled. Further, the term 'very quick runoff' is used for a faster process than 'fast runoff', these terms can be compared with 'overland flow' and 'interflow' respectively. The response times for the very quick runoff, the fast runoff and the groundwater runoff are for most models in the order of 1 day, 8 days and 80 days respectively.

**GR4H-CemaNeige** (Mathevet, 2005) is a combination of the CemaNeige snow module (Valéry et al., 2014) and an hourly version of GR4J (Perrin et al., 2003). GR4H is an empirical four parameters hourly model with a root zone storage and two routing routines: one for very quick and one for fast runoff. The division of water between the two routines is fixed at a 0.1-0.9 ratio; both reservoirs interact with the groundwater. GR4H was developed for high flows rather than for low flows, as low flows are rarely studied at an hourly time step.

**PREvision et Simulations pour l'Annonce et la Gestion des Etiages Sévères, PRESAGES** (Lang et al., 2006), is a daily tool for low flow forecasting and evaluation, but it was slightly modified to run on an hourly time scale. It is derived





from GR4H with differences being no incorporation of interception and snow and a separated groundwater reservoir connected in series with the fast runoff reservoir. There is no longer interaction between the very quick runoff reservoir and the groundwater.

**Wageningen Lowland Runoff Simulator, WALRUS** (Brauer et al., 2014a, b), is a lumped conceptual model for application

in lowland areas with shallow groundwater tables. The model consists of four reservoirs: a combined root zone and groundwater reservoir, a combined very quick and fast runoff reservoir and a surface water reservoir. Snow accumulation and melt are simulated in a pre-processing step. Note that the Ourthe catchment is not located in lowlands; we included WALRUS in the comparison to investigate where a model designed for lowlands would succeed and fail in a hilly catchment.

**M2-M5 models of the SUPERFLEX framework** (Fenicia et al., 2011) are four lumped conceptual models with an increasing degree of complexity. M2 consists of a root zone storage and a non-linear fast runoff reservoir. M3-M5 extend the M2 conceptualisation by adding a lag function (M3-M5), a snow routine (M4-M5) and a groundwater reservoir (M5).

**NAM** is an adapted version of the hydrological model which is coupled to MIKE11 (Nielsen and Hansen, 1973). It consists of a snow module, interception reservoir, root zone storage and a groundwater reservoir; the latter are configured in parallel.

Fast and very quick runoff is generated from the interception reservoir, but depends on the saturation of the root zone storage.

**FLEX-Topo** (Savenije, 2010; Euser et al., 2015) is a conceptual model with three parallel model elements that represent the three dominant hydrological response units (HRUs) of this area: wetlands, hillslopes and agricultural fields. The model elements only interact with each other via the groundwater reservoir and the stream network. Each HRU consists of a

snow module, interception reservoir, root zone storage and fast runoff reservoir. The agricultural area has an additional very quick flow reservoir.

**VHM** (Willems, 2014; Willems et al., 2014) involves a step-wise and data-based procedure to identify a parsimonious lumped conceptual model. For the Ourthe catchment, a model was identified which consists of a root zone storage and three runoff components: very quick runoff, fast runoff and groundwater runoff, which are configured in parallel.

**wflow_hbv** is a completely distributed version of the conceptual HBV model (Lindström et al., 1997) in the wflow framework[3] with a kinematic wave as routing instead of the original triangular routing function. The model has an interception reservoir, snow module, root zone storage, fast runoff reservoir and a groundwater reservoir. The parameter values are constant for the entire catchment, except for maximum interception capacity, which is related to land use.

### 3.3   Evaluation methods

The results of the eleven models and five catchments were compared in multiple ways. First, the scores obtained for the objective functions (NSE and NSElog) were compared. This step enabled to determine the overall performance of the models.

---

[3]http://wflow.readthedocs.io/en/latest/



We expected that this analysis would not reveal much difference; so, two additional analyses were carried out: a statistical analysis and a hydrograph comparison for specific periods.

Three types of statistical analyses and comparisons of simulation results and observations were conducted: cumulative discharges, empirical extreme value distribution of the peak flows and flow duration curves for low flows. The cumulative

discharges were plotted for the entire modelled period and used to investigate the overall water balance. The empirical extreme value distributions were constructed from independent peak discharges, following Willems (2009); the return period was calculated as the mean time interval between the exceedance of given runoff amounts. This analysis of peak flows was carried out to investigate if models were able to simulate the full range of peak discharges observed in the catchments. In addition, the empirical extreme value distribution can provide information on the usefulness of models for flood (frequency) studies and

extrapolations to more extreme events, by assessing the shape of the distribution, as well as the tendency of the difference between higher modelled and observed peak flows. The flow duration curves were constructed for the lowest 20% of the discharges. Low flows are important in the Meuse basin, especially from a user's perspective; comparing observed and simulated flow duration curves helped to assess how well models were able to reproduce low flows.

Finally, specific periods were selected to compare modelled and observed hydrographs. By looking at specific events, more

detailed differences can be observed between models. Four different periods outside the calibration period were selected for this analysis: a summer period, a transition from low to high flows and two winter periods. The summer of 2008 was selected, because many high intensity precipitation events occurred during this period; during the summers in the calibration period, these events did not occur very frequently, making this a benchmark period. The autumn of 2003 was selected as a low to high flow transition period, as 2003 was a very dry summer, so problems with re-saturation were likely to be largest during this year.

The two analysed winter periods were 2002-2003 and 2010. In the studied catchments, winter runoff can consist of rainfall (in 2002-2003) in case of higher temperatures or of snow melt (2010) in case of lower temperatures. By comparing these two winter periods, the model's ability to reproduce both conditions was investigated. It should be noted that not all models contain a snow routine, thus the winter of 2010 was also used to investigate how important a snow routine is for simulating discharges.

## 4   Results

The analyses of metrics, statistics and hydrographs for the eleven model structures, run for the five catchments for the period 1 January 2001 until 31 December 2010, showed different model performances. All analysed figures can be found in the supplement (Sections 3 to 9); overall they confirm that all models perform well (maximum NSE varying between 0.85 and 0.91 and maximum NSElog between 0.85 and 0.93 for the entire modelled period for Tabreux; supplement, Section 2). It was found that even very simple lumped models (M2) could perform as well as very complex (semi-) distributed models

(FLEX-Topo and wflow_hbv) under wet conditions. Most models had higher performances during the validation period than during calibration and blind validation periods in terms of NSE and NSElog, probably due to the wetter conditions during the validation period. The hydrographs and the cumulated discharges over the entire period showed that most models slightly



underestimated observed flows, except for FLEX-Topo. A number of relevant differences between models and catchments are highlighted below.

## 4.1 Internal averaging within the Ourthe catchment

Yearly simulated and observed flows in the Ourthe and its two nested catchments (Ourthe Orientale and Ourthe Occidentale) possibly show the effect of internal averaging, as depicted in Figure 3. While discharged volumes are underestimated by all models in the Ourthe Occidentale, they are overestimated by most models in the Ourthe Orientale and this seems to average out for the Ourthe at Tabreux. Topography, land cover and geology are comparable for the Ourthe Orientale and Ourthe Occidentale catchments, with the Ourthe Orientale catchment being a little steeper and having slightly more forest cover. However, the topography of both catchments differs from that of the Ourthe catchment at Tabreux, indicating that parameters may not be directly regionalised.

Another difference between the Ourthe catchment and its subcatchments is the volume of precipitation and runoff; the Ourthe Orientale catchment receives more precipitation and produces less runoff than the Ourthe Occidentale catchment. Previous studies (e.g., Kleidon and Heimann, 1998; Gao et al., 2014; de Boer-Euser et al., 2016) showed a link between climate (i.e., precipitation and evaporation volumes) and root zone storage capacity. Following their argument, the root zone storage capacity should indeed be larger for the Ourthe Orientale catchment and smaller for the Ourthe Occidentale catchment compared to the Ourthe catchment to meet the evaporative demand of the Ourthe Orientale catchment. Using the root zone storage capacity of the Ourthe catchment for the Ourthe Orientale catchment could lead to too high modelled discharges, using it for the Ourthe Occidentale catchment could lead to too low modelled discharges. On the other hand, it is also possible that precipitation is underestimated in the Ourthe catchment, as all models are underestimating the runoff volume for the Ourthe Occidentale.

## 4.2 Modelling the highest peaks

Figure 4 shows the independent peak flows versus the empirical return period for both observed and modelled discharges for the Lesse. Out of all studied catchments, high flows extremes are the most difficult to capture by the models for the Lesse, making it an interesting case. All models are able to model the lower peaks, but underestimate the higher peaks. This is characterised by two aspects: first, most models do not simulate peaks with the same height as the highest observed peaks. Second, the observations show a steeper increase in peak flows for return periods higher than about 1 year, which most models were not able to capture.

Although these aspects are difficult to simulate, Figure 4 shows that it is not impossible as at least one model (GR4H-CemaNeige) is able to reproduce the steeper increase in peak flow for high return periods and capture the highest peaks. A slight increase was also reproduced by some of the models (PRESAGES, M2, M3, M4, VHM, wflow_hbv). The fact that some models are able to capture the highest peak indicates that data errors and handling may not be the cause of underestimating the highest peaks in these catchments, as one might have concluded if all models had failed.



### 4.3 Modelling low flows

Low flows were analysed by plotting the lowest 20% of the observed and modelled flow duration curves. Discharges during the summer recession periods are generally low (ranging between $0.004 \, \mathrm{mm \, h^{-1}}$ and $0.015 \, \mathrm{mm \, h^{-1}}$ for the lowest 20%) compared to averaged flow ($0.05 \, \mathrm{mm \, h^{-1}}$). The influence of a groundwater reservoir on the modelled discharge is significant as the flow duration curves illustrate, for example for the Ourthe at Tabreux (Figure 5). Adding a groundwater reservoir improves the simulation of the low flows, as illustrated by the difference in performance between models M4 and M5. The only difference between the two models is the presence of a groundwater reservoir and where M4 underestimates low flows, they are properly simulated by M5. This indicates that water is stored during the high flow period in winter and released again during the low flow period in summer.

The configuration of the groundwater reservoir is also important: model structures with a groundwater reservoir parallel to the fast runoff reservoir (M5, NAM, FLEX-Topo, VHM) generally give the best results. Model structures without a groundwater reservoir (M2-M4) underestimate the low flows, while models with a serial or interactive groundwater reservoir (GR4H, PRESAGES, WALRUS, wflow_hbv) overestimate the low flows or model the recessions too steep. On one hand this indicates the importance of preferential recharge in the catchment, on the other hand it indicates the existence of runoff processes with different time scales. With a parallel groundwater reservoir, the time scales for runoff generation are decoupled, with a serial or interactive groundwater reservoir they are connected. The different results for the low flow period indicate that the processes related to the different times scales occur relatively independent in the studied catchments. For the other catchments the same patterns can be found, but slightly shifted upwards or downwards.

### 4.4 The effect of a very quick runoff component

The performance of the models during the 2008 summer was analysed by plotting the hydrographs, as shown in Figure 6. Although total precipitation amounts during this summer were not higher than in other years, the precipitation intensities were. The antecedent root zone storages before the events can be expected to be low, due to high transpiration rates in summer. While most models are not able to capture the summer peaks, VHM and FLEX-Topo are able to simulate the dynamics well, although FLEX-Topo overestimates the summer peaks. As shown in Figure 2, VHM and FLEX-Topo are the only models that contain a very quick runoff component preceding the root zone storage and independent of the root zone storage. Hence, it illustrates that this component is essential for simulating short, intense summer events, which are likely to cause infiltration excess overland flow (i.e., precipitation intensity being higher than infiltration capacity of the soil). Under dry conditions the infiltration capacity of the soil is assumed to be disconnected to the saturation of the soil. Therefore, the very quick flow component should be independent of the root zone storage and should precede it; otherwise these short intense summer rainfall events are stored in the soil instead of discharging directly to the river. Models with a very quick runoff component which is affected by the root zone storage (WALRUS and NAM) and models with a very quick runoff component following the root zone storage (GR4H, wflow_hbv, PRESAGES) do perform better than models where the very quick runoff component is entirely



lacking (M2 to M5), but they miss the sharpness of the response, due to damping of the generated peaks. These findings are consistent for all studied catchments.

## 4.5 Transition from low to high flows

The largest differences in model results between the modelled catchments occur for the transition from low to high flows. For

example, the runoff during the transition from low to high flows between December 2003 and January 2004 are overestimated for all models in the Ourthe Orientale (Figure 7), while only to a minor extent in other catchments. In the Lesse catchment, the performance during this transition period is the highest from the four selected periods for almost all models. In addition, the performance in the Lesse catchment is also higher than that for the Ourthe catchment for almost all models. The variability in performance between models and between subcatchments for this event prevented pinpointing model components that explain

this difference in performance.

Although all models overestimate the discharge of the Ourthe Orientale (Figure 7), their response is different. They especially differ in simulating the two highest peaks: PRESAGES and WALRUS simulate the first one relatively well, but underestimate the second. The other models overestimate the first peak and vary in how they simulate the second one. As the transition period is controlled by the modelled rate of infiltration and its dependence to soil saturation state, one reason could be explained by

differences in modelled evaporation in the antecedent period; however, the model with the lowest evaporation (PRESAGES) is not the model with the highest overestimation of discharge. FLEX-Topo strongly overestimates the discharges; this can partly be caused by the root zone storage capacity. This model has a climate derived root zone storage capacity (de Boer-Euser et al., 2016), which is significantly higher for the Ourthe Orientale catchment than for the Ourthe catchment (see also section 'Internal averaging within the Ourthe catchment'). The difficulty the models encounter to model this transition may be linked

to the hysteretic behaviour in dry-to-wet transition periods (autumn) and in wet-to-dry transition periods (spring).

## 4.6 The effect of a snow routine

The models that include a snow routine (GR4H, WALRUS, M4, M5, NAM, FLEX-Topo and wflow_hbv) did not perform significantly better than the others during snow events. Figure 8 shows the 2010 winter for the Semois catchment: this is the catchment and period with the largest differences between models with and without snow. The models with a snow routine can

reproduce the second highest peak slightly better. The differences are, however, rather small. Although it could be expected that having a snow module improves the performance during a snow event, it was not clearly found in this study. A possible reason for the limited effect of the snow module can be that snow cover mainly occurs for short periods of time. In addition, the discharges corresponding to snow melt periods are similar in magnitude to those originating from liquid precipitation. These two aspects, combined with the use of general metrics for calibration, lead to the possibility that (small) influences of snow on

the discharge are compensated by other parameters when a model does not have a snow module.





## 5 Discussion

### 5.1 Findings about the Meuse basin

The results of this study show first of all differences and similarities between catchments and models. In addition, the analysis of model behaviour under relatively dry conditions (Figures 5 and 6) shows which model configurations are more suitable for these catchments than others: the conceptualisation of the very quick runoff component and of the groundwater reservoir. The very quick runoff component is necessary and should precede infiltration into the root zone storage and not be affected by it. The groundwater reservoir is necessary as well and can best be implemented parallel to fast runoff generation. The effect of a very fast runoff component is directly visible in the hydrographs and consistent for all catchments. The effect of a (different conceptualisation of the) groundwater reservoir is best visible in the lower parts of the flow duration curve and the strength of the effect varies per catchment. We therefore hypothesize that both components are important for a conceptual model of these catchments, especially when the model is aimed to be applicable for analysis of peak and low flows.

These two components are the only two we could generalise for all models and catchments; other results were too variable in space and between model structures and could therefore not be linked to specific model structural components. The comparison consists of eleven model structures, each with specific details. Therefore, other differences and similarities in the modelled discharge could not be easily related to differences and similarities in model conceptualisations.

Another general result of this comparison study is the higher performance of the normal (non-blind) validation period compared to the calibration and blind validation periods. Although performance generally decreases during validation, some studies show an increase in performance (e.g., Hrachowitz et al., 2014) for the validation period. Often, this indicates that hydro-meteorological conditions in the validation period were easier to model. The same holds here: the validation period is the wettest period, and most conceptual models yield the best performance under wet conditions. The higher performance during validation and the hydro-meteorological differences between the calibration and validation periods show that the models are transferable in time and space within our testing protocol.

### 5.2 Benefits of an intercomparison study

Intercomparison studies can provide a more detailed overview of a model's potential than single model studies. In that sense, they enable individual modellers to reflect on familiar model structures, through comparative identification of lacking or relevant components. In a single model study, a poor performance may be easily blamed on data shortcomings or model structural errors. In an intercomparison study, it is less likely that the poor performance of a certain model is due to data errors if there is at least one model that performs well when forced with the same input data.

Preliminary results of the comparative study were sent to the modellers with the question to evaluate the model results and speculate on how their model structure could be improved. One of the responses was that processes were not (or only recently) specifically included in the model (e.g., fast runoff caused by infiltration excess overland flow, snow), because they were not necessary for earlier applications. In addition, the prescribed calibration objectives and lumped precipitation forcing used for most models were brought up as reasons for inferior model performance.





The method used to calibrate VHM for this experiment differed from the normal calibration method applied (Willems, 2009). This may have played an important role in the underestimations of peak flows. As explained by Willems (2009) and others, the NSE objective function applied to the flows at all time steps gives weight to the high flows but less to extremely high flows because they are typically of shorter duration. When comparing the automatic calibration applied for this study to the manual

calibration normally applied, focussing on the hydrological extremes (Willems, 2014), improved results for peak flows are obtained for the manual calibration, as illustrated in Figure 9.

This intercomparison study shows that the assessed models have different strengths in capturing specific characteristics of the runoff response. Single models may have been developed to perform better on a specific aspect at the expense of another one, as explained by Duan et al. (2007). Applying a multimodel ensemble instead of relying on a single model outcome provides

more information on model structure uncertainty. This helps hydrologists to better understand the catchment functioning and improve uncertainty estimations. In an operational context, multimodel ensemble are useful to make more informed decisions.

### 5.3  Comparison of models

The choice of calibration method was left to the individual modellers, with the only constraint that the Nash-Sutcliffe Efficiency of the discharge (NSE) and of the log of the discharge (NSElog) had to be used as objective functions. This resulted in some

modellers using a search algorithm, while others applied uniform sampling of the parameter space. In addition, the (width of the) parameter space before sampling varied per model. This freedom in calibration probably has affected the results; on the other hand, we considered that the calibration algorithm chosen is strongly linked to the model and modeller's experience. As some methods used a search algorithm while others applied uniform sampling, the range of the model realisations varied considerably between models: for some models the 20 realisations were almost identical, while for others there were large

differences. This added an additional source of variability to the comparison, but this variability did not alter the conclusions.

After the calibration on NSE and NSElog, the models were compared focussing on specific periods and statistics. Although the general metrics showed a high performance for all models, (large) differences are observed when focusing on the specific periods or statistics. This is especially true when modelling events under drier conditions, which are the conditions when different model behaviors were most visible. A model evaluation based on visual inspection of the hydrographs during specific

events may sound subjective, but because it focuses on very specific events, the human eye easily detects patterns that reflect model performance. This emphasises again the importance of a broad but specific model evaluation, especially for a model comparison study.

The majority of the models considered in this study is lumped and used lumped forcing. Only two models, FLEX-Topo and wflow_hbv, used the semi and completely distributed forcing respectively. The distribution of the forcing and the model did

not seem to have a significant impact on model performance compared to the other models. The differences in model structure affected model performance more than the differences in distribution of forcing. This is in line with earlier studies (e.g., Euser et al., 2015; Vansteenkiste et al., 2014), which showed that distribution of forcing data has a smaller effect on performance than the selection of model structure.





The varying degree of experience of the modellers with both their model and calibration technique and with the studied areas is likely to influence the reproducibility of this experiment. However, the similar forcing data used and the defined protocol enabled to reduce the degrees of freedom of the modellers and enabled the comparison of the results. This study is a large step forward in the international cooperation between universities and institutes working in the Meuse basin. Sharing data

and model results in this set-up has never been realised before, but it is fundamental to open up the dialogue and advance hydrological understanding of the studied catchment in a more coherent way.

## 5.4 Future intercomparison studies

We think that international model intercomparison studies are very important and are definitely valuable in future research programs. First of all, they are a good opportunity to increase cooperation and discussion between different institutes. In

addition, it is a good means for young scientists to get to know the models used in neighbouring universities and institutes.

To increase the possibility to draw strong conclusions about the hydrological functioning of a catchment, a different set-up may be useful. If all modellers would select a very strong element of their model, this could be incorporated in all the other models. By doing this, in a controlled sequence and actually creating a virtual laboratory, probably more insight could be obtained regarding hydrological functioning and suitable model conceptualisations. In addition, more independent data sources,

besides discharge, would probably strongly increase the possibility to obtain insight about the hydrological functioning of the studied catchments.

## 6 Conclusions

For this study we compared eleven models for five subcatchments of the Meuse basin. All models were calibrated on the Ourthe at Tabreux; they were then evaluated for two different periods and five different catchments. NSE values for all models and all

catchments were comparable, with in some cases even higher performances during the validation period. Although NSE values were comparable, a more detailed analysis, focussing on specific events through hydrograph inspection and statistics, revealed clear differences between the models, especially for drier conditions. We found that a very quick runoff component preceding and not affected by the unsaturated store was relevant to model the hydrological response after short and intense summer events. This conceptualisation ensures that water is not stored in the soil but quickly flows to the river. Also a groundwater

reservoir implemented in parallel to the fast runoff generation seemed necessary to model the recession best. Thus, from this study we can conclude that often more detailed analyses are required to relate differences in the hydrograph to model structure components. A model intercomparison study is a valuable approach to draw conclusions about hydrological functioning of a system, and most of all, it is a great opportunity to reflect on your model structure by comparing it with other models. This leads to the question: "What is my model doing well in comparison to other models and why?". This points out the model

structure components to keep and in the end, focussing on this question will improve our hydrological understanding.



*Acknowledgements.* The authors would like to thank the Service Public de Wallonie, Direction générale opérationnelle de la Mobilité et des Voies hydrauliques, Département des Etudes et de l'Appui à la Gestion, Direction de la Gestion hydrologique intégrée (Bld du Nord 8-5000 Namur, Belgium) for providing the precipitation and discharge data. We would like to thank Bernhard Becker for organising the Meuse symposia that have led to this fruitful cooperation and thank Frederiek Sperna Weiland for the valuable discussions and her contribution in

5  data preparation. Further we would like to thank Michel Pirotton for his review and Pierre Archambeau, Sébastien Erpicum, Michel Pirotton for the analysis of the impact of the Nisramont dam presented in the supplement.



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





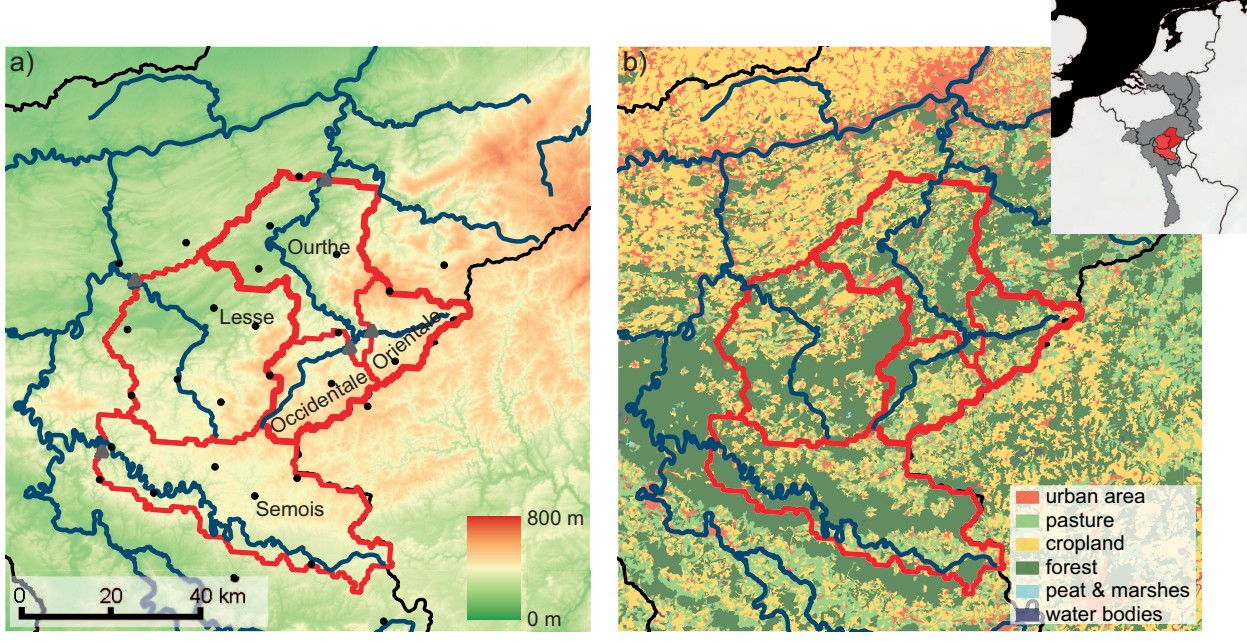

**Figure 1.** a) Studied catchments (red outline) with DEM and used rain (black dots) and discharge (grey triangles) gauges; b) studied catchments with land cover. The thin black line indicates the catchment boundary of the Meuse River. DEM is obtained from http://hydrosheds.cr.usgs.gov, on 05-06-2013; land use data is derived from CORINE Land Cover (European Environmental Agency, 2006).

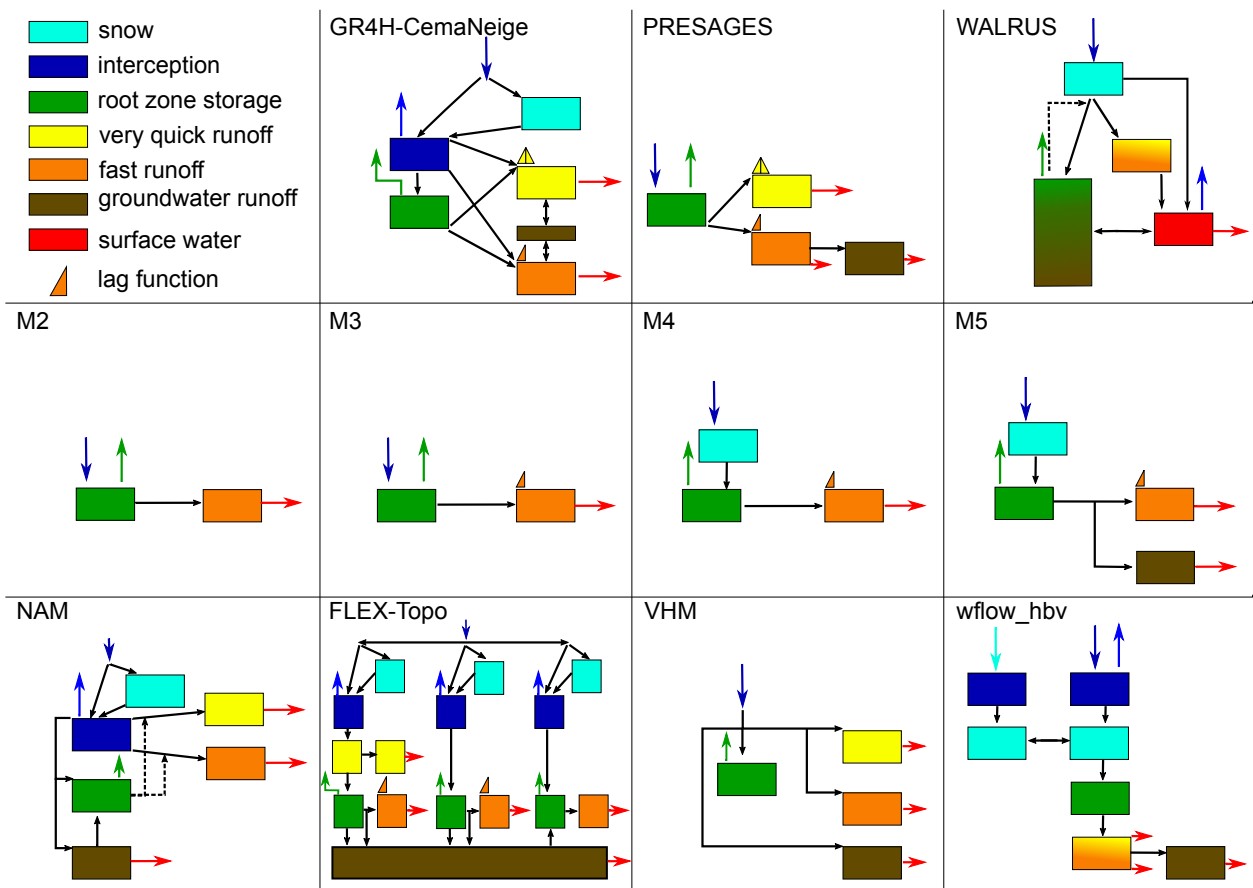

**Figure 2.** Overview of the eleven used model structures: the schematisation of the model structures is slightly simplified, with the aim to highlight the similarities and differences between the models. The solid lines indicate fluxes between model storages; the dashed lines indicate the influence of the state in a model storage to a flux.





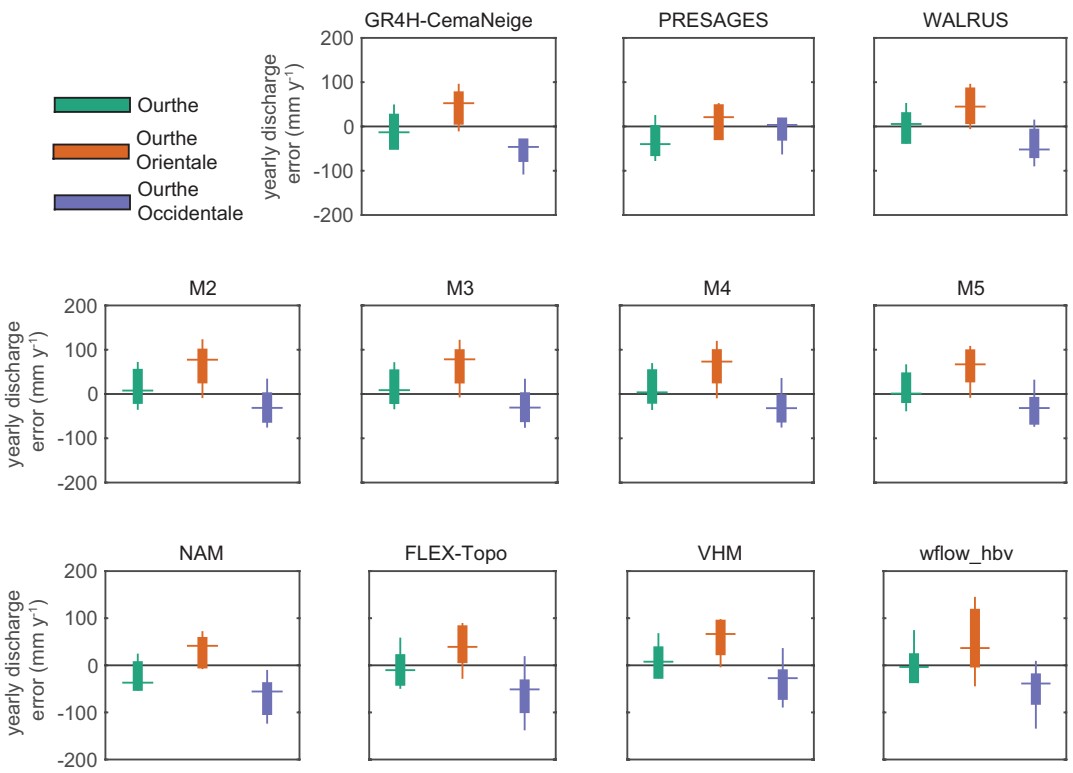

**Figure 3.** Difference between observed and modelled yearly discharge for Ourthe (green bars), Ourthe Orientale (orange bars) and Ourthe Occidentale (purple bars). Note: to make the graphs more readable, outliers were not plotted





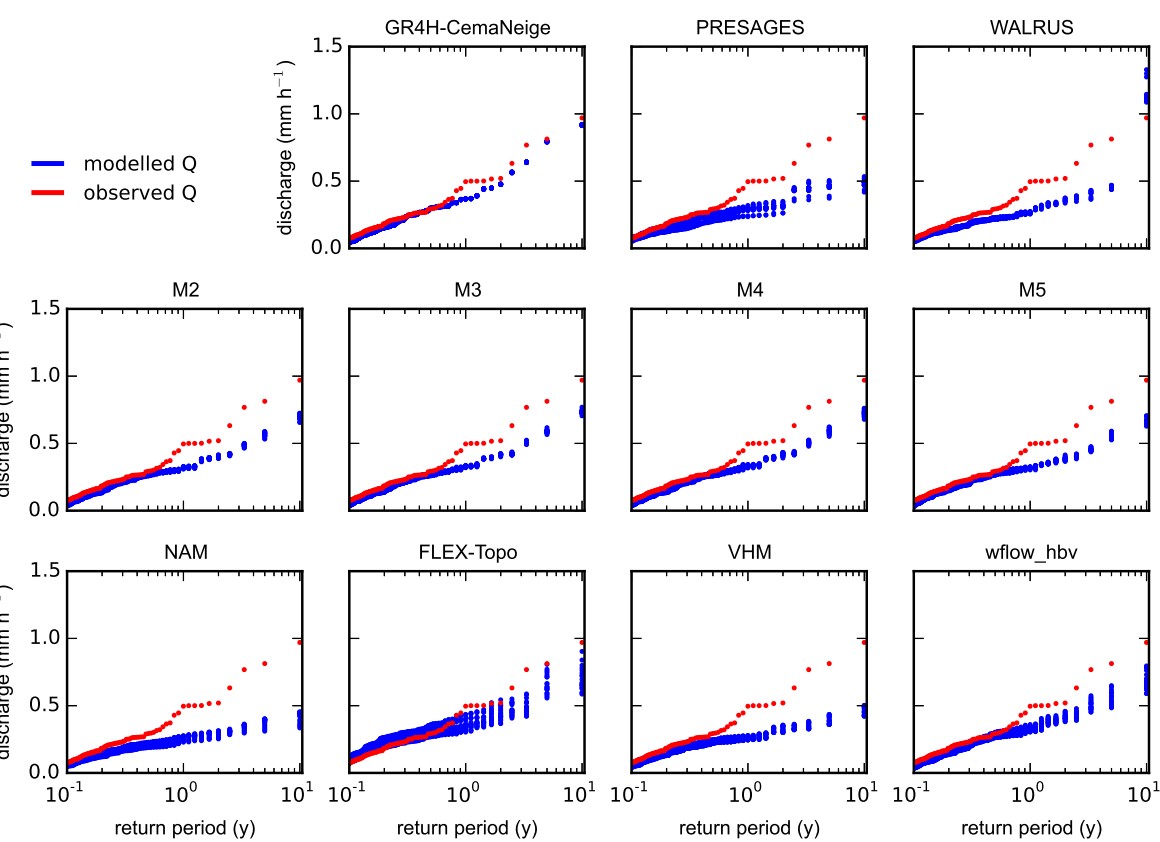

**Figure 4.** Empirical extreme value analysis for peak flows for the Lesse at Gendron for the total modelled period (2001-2010) (red dots = observed, blue dots = modelled, the spread in the blue dots shows the different realisations)





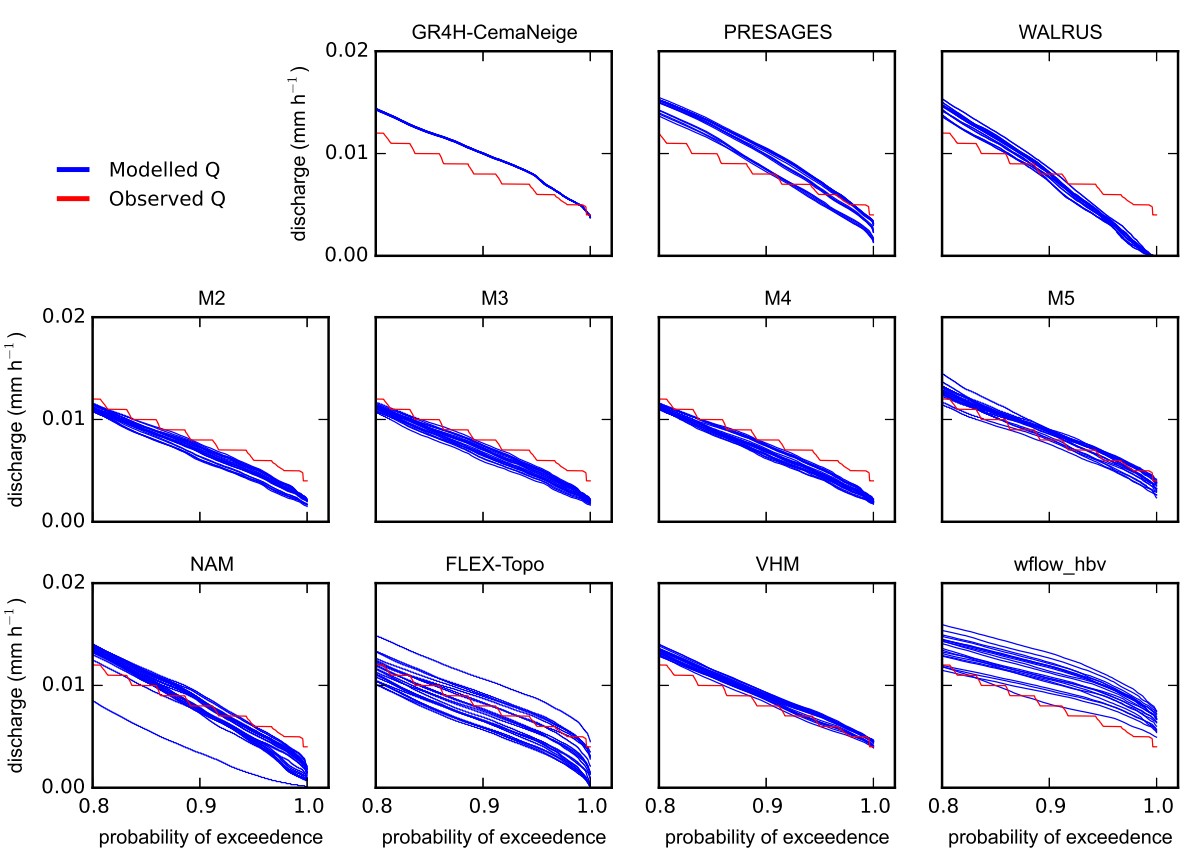

**Figure 5.** Lowest 20% of the flow duration curves for the Ourthe at Tabreux for all models (red line = observed, blue lines = modelled).

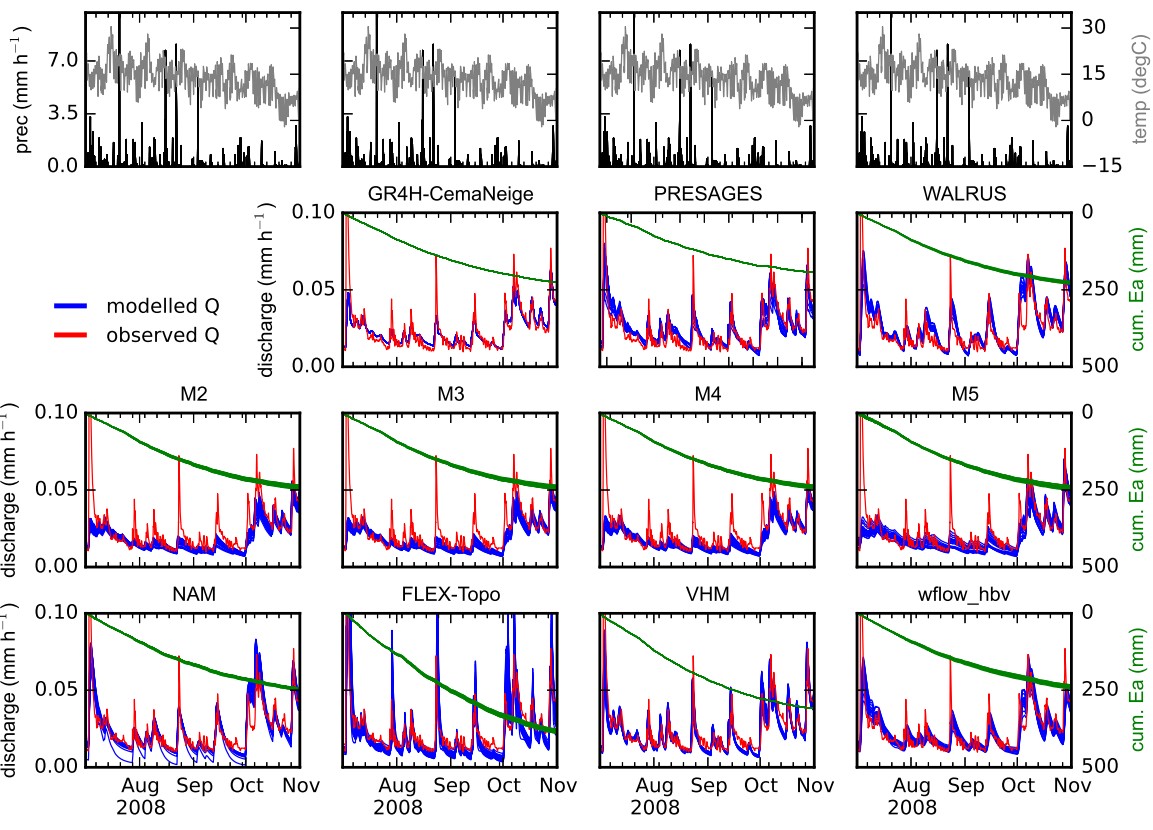

**Figure 6.** Modelled (blue) and observed (red) discharges for summer 2008 for the Ourthe at Tabreux. The green line shows the cumulative actual evaporation for the plotted period. Note: the four graphs with precipitation and temperature on top are the same.





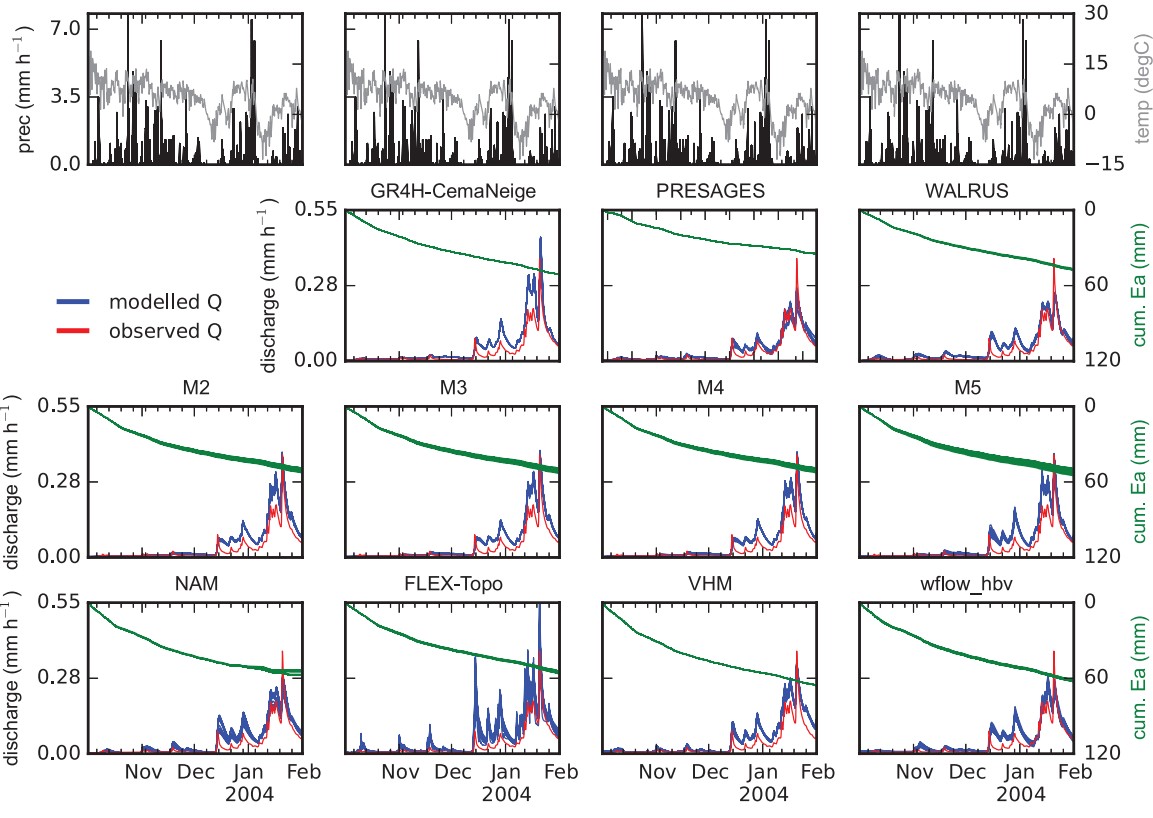

**Figure 7.** Modelled (blue) and observed (red) discharges for autumn 2003 for the Ourthe Orientale at Mabompré. The green line shows the cumulative actual evaporation for the plotted period. Note: the four graphs with precipitation and potential evaporation on top are the same.





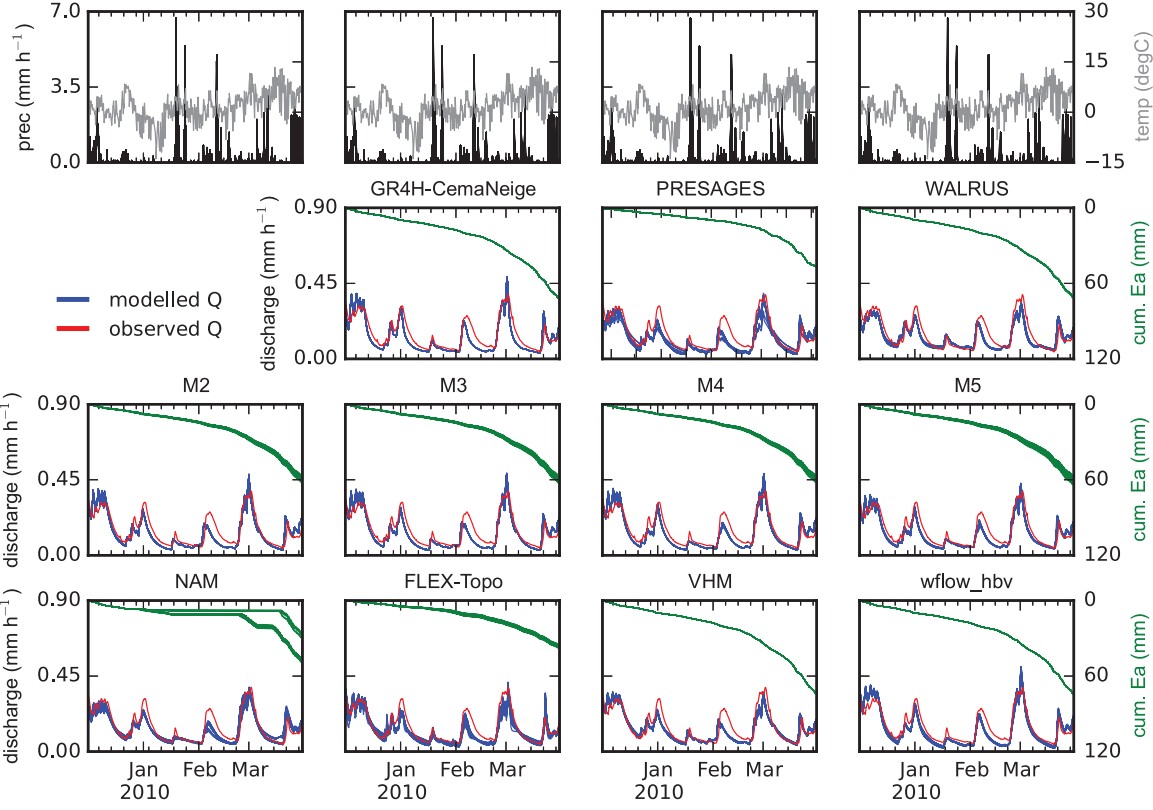

**Figure 8.** Modelled (blue) and observed (red) discharge for the Semois at Membre for the 2010 winter period. The green line shows the cumulative actual evaporation for the plotted period. Note that the plots with precipitation and temperature on top are the same.

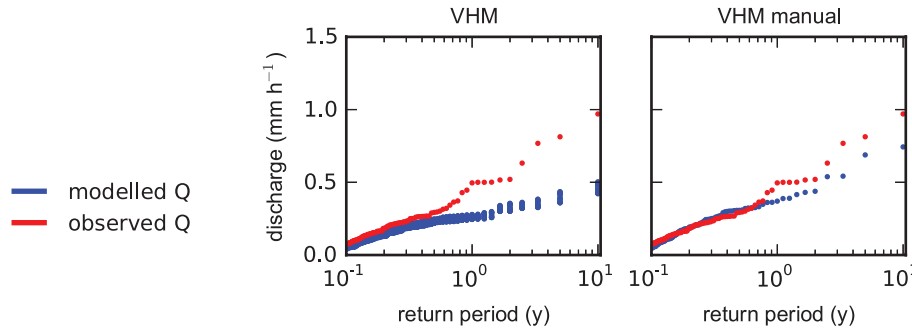

**Figure 9.** Difference between observed (red) and modelled (blue) empirical frequency distribution of peak flows for the Lesse at Gendron for the entire modelled period (2001-2010) for the automatic (left) and the manual (right) VHM calibrations





**Table 1.** Catchment characteristics

|  | Ourthe | Orientale | Occidentale | Semois | Lesse |
|---|---|---|---|---|---|
| Catchment area (km$^2$) | 1607 | 317 | 379 | 1226 | 1286 |
| Max elevation (m) | 662 | 662 | 596 | 569 | 586 |
| Min elevation (m) | 108 | 294 | 303 | 176 | 116 |
| Elevation range (m) | 554 | 368 | 293 | 393 | 482 |
| Mean slope | 0.090 | 0.081 | 0.077 | 0.087 | 0.086 |
| Max slope | 0.75 | 0.62 | 0.58 | 0.94 | 0.79 |
| Max flow distance (km) | 144 | 32 | 44 | 174 | 83 |
| Forest cover (%) | 46 | 48 | 40 | 56 | 55 |
| Pasture cover (%) | 21 | 20 | 23 | 18 | 11 |
| Urban cover (%) | 6 | 5 | 4 | 5 | 5 |
| Crop cover (%) | 27 | 27 | 33 | 21 | 29 |
| Mean annual precipitation (mm y$^{-1}$) | 1000 | 1080 | 1010 | 1250 | 1000 |
| Mean annual runoff (mm y$^{-1}$) | 460 | 480 | 500 | 670 | 420 |
| Mean annual temperature (°C) | 9.6 | 9.1 | 9.3 | 9.6 | 9.8 |
| Mean annual pot evaporation (mm y$^{-1}$) | 730 | 710 | 720 | 750 | 740 |





**Table 2.** Characteristics of the configuration of the different models

| Model | Forcing | Calibration | Regionalisation | Group |
|---|---|---|---|---|
| GR4H | Lumped | Pre-filtering of parameter space using three quantiles for each of the four parameters, followed by stepwise calibration to optimum | No | IRSTEA |
| PRESAGES | Lumped | Optimization with 100 starting points within the parameter space that converge to local minima, which results in more than 2000 parameter sets | River routing based on catchment area | Université de Lorraine |
| WALRUS | Lumped | Manual narrowing of parameter space 500 samples with latin hypercube, 10 best ones for Levenberg-Marquardt optimisation | No | Wageningen University |
| M2 | Lumped | MOSCEM-UA (Vrugt et al., 2003) | No | Eawag |
| M3 | Lumped | MOSCEM-UA | No | Eawag |
| M4 | Lumped | MOSCEM-UA | No | Eawag |
| M5 | Lumped | MOSCEM-UA | No | Eawag |
| NAM | Lumped | DREAM_ZS (Laloy and Vrugt, 2012) | No | Vrije Universiteit Brussel |
| FLEX-Topo | Semi-distributed | Manual narrowing of parameter space, 2000 uniform samples | Percentages HRUs; hydraulic length | Delft University of Technology |
| VHM | Lumped | MOSCEM-UA | No | KU Leuven |
| Wflow_hbv | Distributed | Manual narrowing of parameter space, 2000 uniform samples | Interception capacity | Deltares |