# Peer review of "Looking beyond general metrics for model comparison - lessons from an international model intercomparison study"

_Hydrology and Earth System Sciences, 2016_

## Referee Comment (RC1) · Anonymous Referee #1 · 3 Oct 2016

It is a well written and interesting paper, which addresses relevant scientific questions within the scope of HESS. The concepts for model comparison are novel. There are substantial conclusions, about the need for including processes (quick flow reservoir/parallel groundwater storage), based on comparison of the different conceptual rainfall-runoff models, tested to for the same catchment / data. Scientific methods are valid and clearly outlined. However, even though scientific significance and quality in my view is excellent, when it comes to presentation quality, I would recommend minor improvements, before the paper is published. 1) It is a bit unclear if this is a comparative study or a model comparison (or intercomparison) of different process descriptions. The paper uses different terminology of comparative in different Places. Please check

and make clear that it is consistent throughout the paper. 2) How has the problem definition (setting objectives) for the rainfall-runoff model been 'stated'. When reading the paper, it seems like an overall objective has been to test different conceptual models overall performance for both high flows and low flows at the same time, as defined by the metrics (NSU + NSUlog transformed) and the qualitative high, low and transition flow performance. However, in the mission statement, how was this described for the different modelteams/universities participating in the study (the paper is a bit vague on this issue)? I miss a bit on this, was the approach for qualitative testing of the different models know, or not know by the 'modellers'? I guess the model study objectives and the described objectives of the paper (paper 3, last 5 lines of section 1) may not have been fully the same, so I suggest that this 'mission statement' for carrying out the calibration and validation of each conceptual rainfall runoff model, should be briefly mentioned with a few lines, before the objectives of the paper is described. 3) The comparison based on quantitative metrics (NSU and NSUlog) and qualitative criteria (low flow duration curves, extreme value distribution plots, simulated timeseries for low flow, timeseries for transition from low to high flow and timeseries for winter periods), basically reveals that NSU and NSUlog cannot discriminate between different conceptual models. In stead, events is introduced , which then is used for arguing that the need for including a very quick flow reservoir preceding the root zone storage, and a slow reservoir in parallel with the fast reservoir to model the recession for the Ourthe catchment. This is fine, but, it is a very subjective evaluation, since there is no metrics for how to evaluate which model gives the best results in the selected events (eg. Summer 2008 etc.). Why have you not calculated some statistics for these events, in order to measure in operational way, the differences in performance of the different models? 1-3) As a general suggestion for improving the paper I would recommend a Figure as part of the section on Methods or in the discussion which clearly illustrate the 'modelling protocol' (or suggestion for such after the study) and how the study went through the various steps in a rainfall-runoff modelling process (e.g. a workflow of the modelling process from define purpose, conceptual model building, setup of numerical

models to split-sample/proxy basin tests). Specific comments: Abstract. OK! 1.Introduction Well written. Ok. 2.Study areas and data Since the forestry part is high (46 %), it could be relevant to unfold a bit more how interception storage is addressed in the various conceptual models. 3.Methods Please add 1-2 references on page 5/line 6 to NSElog. Would it be possible to add to the description of each model the number of parameters which requires calibration? (did any modelling teams do sensitivity analysis before selection calibration parameters, if not calibrating on all parameters?) Either in the text or in Table 2. 4. Results Please add a table with the results of NSE and NSElog for each calibration and validation period for each station (not the min/max but the average NSE and NSElog for the period). I don't think that it is fully satisfying for the reader to have to consult the background, supplementary material. 5. Discussion and 6. Conclusion OK.

---

## Referee Comment (RC2) · Anonymous Referee #2 · 5 Oct 2016

Overall The paper summerises the outcome of a model intercomparison exercise, using 11 models in one catchment with performance evaluation both in time and space. The experiment seems well performed and the paper is well written. In general, model inter-comparisons should be encouraged to learn more about different model concepts, increase transparency in science and identify appropriate process descriptions. This paper is a good role-model for how such studies should be performed and documented. It is especially important as it considers a transboundary river, which is probably exposed to many model approaches resulting in different model results – this might be very confusing to decision-makers and lead to disputes across the borders.

The paper is based on state-of the-art modelling practices and presents a neat study

in a clear and concise way. It is ambitious in linking hydrological understanding to the differences in model performance. However, I think the authors should try to extract even more knowledge about processes controlling runoff at various stages of the hydrograph in the catchment as well as identify knowledge gaps, based on the ensemble modelling experiment. I also think it is important to highlight that even though the models showed on average good model performance, some parts of the hydrograph is still poorly described and should be identified. This is looking beyond what is available in general metrics! I suggest publication in HESS with only minor changes.

I have added a few detailed comments, which may improve the paper even further:

Abstract: I think the abstract can be a bit longer. I miss this sentence from the Methods section to describe the analysis made in the experiment "Three types of statistical analyses and comparisons of simulation results and observations were conducted: cumulative discharges, empirical extreme value distribution of the peak flows and flow duration curves for low flows." I would also recommend highlighting in the abstract more findings when it comes to understanding controls of these parts of the hydrograph, related to differences in model performance, as well as, the fact that some parts of the hydrograph is poorly understood and here we can thus identify present knowledge gaps.

Introduction The first part is a bit too general and has many references to the literature, which are not necessary for the understanding of the paper and as a reader I miss guidance to how the citied papers actually contributed to the topics discussed. Often the references are lumped and support general statements that doesn't help the reader much in deeper understanding of the literature and previous work, see example below.

Page 2, row 12: reading "Hydrological studies at different scales and under different climates have shown a large variety of hypotheses on hydrological functioning (e.g., McDonnell, 2013; Zehe et al., 2013; Fenicia et al., 2013; Clark et al., 2016; Seibert et al., 2016)." This is an example of a very general statement with no guidance to what

hypothesis the references refer to. Please, give examples and divide the lumped chain of references to explain to the author why these references are chosen – what variety did they show in hydrological understanding?

Page 2 row 22: I suggest starting the Introduction here – the paragraphs above don't contribute much but are common knowledge. This is where the study is motivated and from this point the text is much more interesting and straightforward. (Avoid reference dropping!)

Page 2 row 31: This sentence is difficult to understand: Ceola et al. (2015) concluded that deriving the causes of performance differences between various model structures is not trivial, mainly due to the considerable differences in model structures which disturbs the identification of model features that increase model performance. Are you trying to explain the problem of equifinality here? (i.e. that many different parameter settings may result in similar model performance, due to compensating processes in the model description?) Please, rephrase!

Page 4 row 15: I guess the notation of 'local time' can be removed in this decadal context.

Method The experiment is very straight forward and described with relevant level of details. I like the schematic pictures of model structures of Fig 2 and the descriptions in Table 2. Very useful for understanding!

Results I suggest including statistical metrics for model performance seen in Fig 3-9. Each graph can be assigned with a (few) metric(s) for the data it is showing, to support the analysis.

Section 4.2 Modelling the highest peaks: please, elaborate on potential causes why some models are more successful than other to capture the peaks. Are there processes they did describe that others did not? Where they more carefully calibrated? (did they apply other considerations for parameter choices?)

Fig 3: it would be more interesting to see relative error of annual flow instead of absolute; at least for a reader who is not familiar with the catchment but with general model performances.

Section 4.5 Transition from low to high flows – what can be learnt from these results of poor model performance? Interesting that a snow routine was not necessary, but could be compensated for... how?

Discussion Section 5.1: With the title 'Findings about the Meuse basin' I suggest you add some findings about which processes that seem to control flow of high peaks, low flow, etc. based on your results from model performance using different process descriptions/structures. Alternatively, you should change the title to "Model performance in the Meuse basin" but this will make the paper less appealing in my view. I would rather like to see the hydrological interpretation from the model results (i.e. change perspective to describe nature instead of models).

Page 11, row 10: I suggest to rephrase "We therefore hypothesize..." to We therefore suggest...or rather: " The results thus indicate...". Even if results may raise new questions, I think you should take the opportunity to make a statement from your study here.

Section 5.2 Benefits of an intercomparison study: Please, specify more clearly what direct benefit you could draw from using a model ensemble in this catchment. It is a costly and time-consuming process, so what are the proofs from your results claiming that it is worthwhile?

Page 11, row 28: I suggest acknowledging the importance of following a firm protocol (as you seem to have done here) in the collaboration, to ensure transparency for the result analysis and reproducibility of computational experiments. See for instance: Hutton, C., Wagener, T., Freer, J., Han, D., Duffy, C. and Arheimer, B., 2016. Most computational hydrology is not reproducible, so is it really science?. Water Resour. Res.. Accepted Author Manuscript. doi:10.1002/2016WR019285

Page 12 row 6: Note that it is quite common that catchment models underestimate extremes even though they do reasonable good on mean values, also in a multi-basin analysis. See for instance: Donnelly, C, Andersson, J.C.M. and Arheimer, B., 2016. Using flow signatures and catchment similarities to evaluate a multi-basin model (E-HYPE) across Europe. Hydr. Sciences Journal 61(2):255-273, doi: 10.1080/02626667.2015.1027710

Conclusion I would recommend highlighting your findings in understanding controls of the various parts of the hydrograph, related to differences in model performance. Was it only events during dry conditions you could refer to improved processes understanding? Check if there is more from your work to extract here! For 'Transition from low to high flows' as well as extreme flows I think you should acknowledge that we see knowledge gaps in process understanding and identify weaknesses in the model descriptions.
* * *

---

## Referee Comment (RC3) · R. Kumar (Referee) · 7 Oct 2016

This paper by Boer-Euser et al describes the modeling inter-comparison results undertaken by a group of modelers over the Ourthe catchment in Belgium. They follow a fixed set of protocols for their model set-up; and the results shows the benefit of using event-specific metrics for understanding intermodal differences. The paper is quite well written and in my opinion it fits quite wells within the scope of the HESS journal. In the following, I provide some general/specific comments, which the authors may consider while revising the manuscript.

1. Missing references (relevant to this study) - specifically of the distributed model inter-comparison studies (phase 1 and 2), which have also analyzed model skill using

event specific metrics for specifying inter-model differences.

Reed, S., Koren, V., Smith, M., Zhang, Z., Moreda, F., Seo, D. J., & Participants, D. M. I. P. (2004). Overall distributed model intercomparison project results. Journal of Hydrology, 298(1), 27-60.

Smith, M. B., Koren, V., Zhang, Z., Zhang, Y., Reed, S. M., Cui, Z., ... & Anderson, E. A. (2012). Results of the DMIP 2 Oklahoma experiments. Journal of Hydrology, 418, 17-48.

2. I would suggest the authors to illustrate the modeling protocol in a step-wise manner or a flow chart to better follow the content.

3. It was bit difficult for me to follow the results - switching from one catchment to another. The rational/reason behind presenting results in this way should be clear to the readers.

4. I would also suggest the authors to comprehensively present their results for different catchments in a tabular form.

5. Blind validation in space: How different are the selected catchments in terms of their (dynamic) hydrologic behaviors? - for example in terms of correlation metrics of daily/hourly stream flows -

6. Among all the selected models, the GR4H model appears to have the least (or almost no) uncertainty in model outputs. Please explain?

7. Section 4.2 "Modelling the highest peaks" could be revised as "Modelling the flood peaks"

Good luck with the Revision

Rohini Kumar
* * *

---

## Author Comment (AC3) · 11 Oct 2016

Dear Rohini Kumar,

Thank you for your positive evaluation of our manuscript. We value the comments and suggestions you have made and would like to respond to them below.

*1. Missing references (relevant to this study) - specifically of the distributed model inter-comparison studies (phase 1 and 2), which have also analyzed model skill using event specific metrics for specifying inter-model differences.*
*Reed, S., Koren, V., Smith, M., Zhang, Z., Moreda, F., Seo, D. J., Participants, D. M. I. P. (2004). Overall distributed model intercomparison project results. Journal of*

*Hydrology, 298(1), 27-60.*

*Smith, M. B., Koren, V., Zhang, Z., Zhang, Y., Reed, S. M., Cui, Z., ... Anderson, E. A. (2012). Results of the DMIP 2 Oklahoma experiments. Journal of Hydrology, 418, 17-48.*

Thank you for pointing out these two related studies, we will consult them and include them in the revised version of the manuscript.

*2. I would suggest the authors to illustrate the modeling protocol in a step-wise manner or a flow chart to better follow the content.*

We agree with you that a flow chart with the modelling protocol would make it easier to follow for the reader. In our reply to the first reviewer, we proposed a possible schematisation for this chart.

*3. It was bit difficult for me to follow the results - switching from one catchment to another. The rational/reason behind presenting results in this way should be clear to the readers.*

We understand that currently the selection of presented results might seem a bit arbitrary. However, in each section of the results, we selected a catchment that illustrated most clearly the message we wanted to convey. We will make sure that the reason for choosing one catchment over the other is clearly stated in the text. The results for all catchments are shown in the supplement to support the selection made for the manuscript itself.

*4. I would also suggest the authors to comprehensively present their results for different catchments in a tabular form.*

We agree with you that in certain cases a table can be very informative to present results. However, we do not think that a table with NSE/NSElog values for all catchments will be very informative, as we also pointed out in our reply to the first reviewer. We could try to make a table which qualitatively summarises the results we describe in the

different paragraphs of section 4 for all catchments, but this table has as well a high risk of being very large and therefore difficult to interpret.

*5. Blind validation in space: How different are the selected catchments in terms of their (dynamic) hydrologic behaviors? - for example in terms of correlation metrics of daily/hourly stream flows -*

Thank you for asking this question. In general, the hydrological behaviour of the catchments is similar, we will provide some more details about the daily/hourly stream flows in the revised version of the manuscript; either in the text or in Table 1.

*6. Among all the selected models, the GR4H model appears to have the least (or almost no) uncertainty in model outputs. Please explain?*

The band width in the 20 model realisations of GR4H is indeed very narrow. This is due to the calibration algorithm applied, which followed a stepwise calibration to an optimum after pre-filtering of the parameter space (see Table 2). This aspect of the results was discussed in section 5.3, however we will point this out in the result section as well.

*7. Section 4.2 "Modelling the highest peaks" could be revised as "Modelling the flood peaks"*

Thank you for pointing this out, we will change the title of this section.

*Good luck with the Revision Rohini Kumar*

Thank you!

---

## Author Response (AR1)

Dear editor,

Thank you for the positive evaluation of our manuscript. We have incorporated the points raised by the reviewers. Especially, we added metrics to the graphs in the result section and added a figure in which we summarise the results of all the models and catchments for a set of signatures.

This document contains the replies to the reviewers. These replies are slightly different from the ones posted in the discussion, because during the revision we decided to incorporate certain comments in a slightly different way. Following the replies to the reviewers, this document also contains a version of the manuscript with all changes highlighted. In the supplement the same metrics were added to plots as in the paper.

We have uploaded the revised abstract, manuscript and supplement without the changes highlighted as separate files.

On behave of all authors,

Kind regards,
Tanja de Boer-Euser

**Reply to review of Anonymous referee #1**

Dear reviewer,

Thank you for your positive evaluation of our manuscript. We appreciate the comments and suggestions you have made and would like to respond to them below.

1. *It is a bit unclear if this is a comparative study or a model comparison (or intercomparison) of different process descriptions. The paper uses different terminology of comparative in different Places. Please check and make clear that it is consistent throughout the paper.*

   Thank you for this comment, we fully agree with you that the used terminology should be consistent throughout the paper. The paper is meant as a model intercomparison and as such investigates whether certain processes are more suitable to apply in models than others. We have made this consistent in the revised version of the manuscript.

2. *How has the problem definition (setting objectives) for the rainfall-runoff model been stated. When reading the paper, it seems like an overall objective has been to test different conceptual models overall performance for both high flows and low flows at the same time, as defined by the metrics (NSU + NSUlog transformed) and the qualitative high, low and transition flow performance. However, in the mission statement, how was this described for the different model teams/universities participating in the study (the paper is a bit vague on this issue)? I miss a bit on this, was the approach for qualitative testing of the different models know, or not know by the modellers? I guess the model study objectives and the described objectives of the paper (paper 3, last 5 lines of section 1) may not have been fully the same, so I suggest that this mission statement for carrying out the calibration and validation of each conceptual rainfall runoff model, should be briefly mentioned with a few lines, before the objectives of the paper is described.*

   Thank you for this comment, we realise that the current phrasing may be a bit confusing. The objective of the study was to force a set of models, used by different teams in the Meuse basin, with the same meteorological input and to compare the model outputs. To do this, we needed to specify two elements in more detail: 1) how should the models be calibrated; 2) how should the model outputs be compared (e.g., model evaluation tools). For the first one we decided that modellers probably know best how to calibrate their own model, so we only set the objective functions (NSE and NSElog) and the number of desired model realisations (20). For the second, we decided to base the comparison not only on the calibration criteria, but also on aspects that were not specifically taken into account during the calibration procedure, so as to investigate the full range of a models capabilities. As the data was of good quality and the models had already proven their usefulness, we were not surprised that the NSE and NSElog values did not discriminate much between the models. However, this makes the other findings even more interesting. We have added a figure to the manuscript and made the distinction between calibration objectives and model evaluation tools more clear in the revised version of the manuscript.

3. *The comparison based on quantitative metrics (NSE and NSElog) and qualitative criteria (low flow duration curves, extreme value distribution plots, simulated timeseries for low flow, timeseries for transition from low to high flow and timeseries for winter periods), basically reveals that NSE and NSElog cannot discriminate between different conceptual models. Instead, events is introduced, which then is used for arguing that the need for including a very quick flow reservoir preceding the root zone storage, and a slow reservoir in parallel with the fast reservoir to model the recession for the Ourthe catchment. This is fine, but, it is a very subjective evaluation, since there is no metrics for how to evaluate which model gives the best results in the selected events (eg. Summer 2008 etc.). Why have you not calculated some statistics for these events, in order to measure in operational way, the differences in performance of the different models?*

   In the different steps of comparing the model output we did calculate a range of statistics and hydrological signatures. However, the main problem was that a certain metric only quantifies a specific element of the performance of a model. So, to make a fair comparison between the models (each models excels on other elements), a whole range of metrics was required. Following on this, the more metrics are used, the more difficult it is to present the results in a clear and easy to interpret way to the reader. Therefore, we used the human eye as evaluator in the first version of the manuscript. However, we agree with you that it makes the evaluation subjective. Thus, in the revised version we have added some metrics to quantify our observations for the different events.

4. *As a general suggestion for improving the paper I would recommend a Figure as part of the section on Methods or in the discussion which clearly illustrate the 'modelling protocol' (or suggestion for such after the study) and how the study went through the various steps in a rainfall-runoff modelling process (e.g. a workflow of the modelling process from define purpose, conceptual model building, setup of numerical models to split-sample/proxy basin tests).*

   Thank you for the suggestion. We have added such a figure.

**Specific comments:**

2.Study areas and data

*Since the forestry part is high (46 %), it could be relevant to unfold a bit more how interception storage is addressed in the various conceptual models.*

We agree and we have briefly discussed the differences in interception module between the models in the revised version of the manuscript.

3.Methods

*Please add 1-2 references on page 5/line 6 to NSElog. Would it be possible to add to the description of each model the number of parameters which requires calibration? (did any modelling teams do sensitivity analysis before selection calibration parameters, if not calibrating on all parameters?) Either in the text or in Table 2.*

We have added a references here in which NSElog is used as a performance indicator for low

flows. Regarding the number of parameters that were calibrated for each model, we have added them to Table 2. Regarding the sensitivity analysis, most of the calibration methods that used pre-filtering of the parameter space implicitly used a sensitivity analysis.

4. Results

*Please add a table with the results of NSE and NSElog for each calibration and validation period for each station (not the min/max but the average NSE and NSElog for the period). I dont think that it is fully satisfying for the reader to have to consult the background, supplementary material.*

Here we do not fully agree: the suggested table would be too large (2 metrics * 5 stations * 3 periods * 11 models) and would mainly contain very similar numbers. As we do not think that such a table is informative to the reader, we prefer to keep the table(s) in the supplement. However, we have added an additional figure to the revised manuscript in which the results for all catchments and all models are summarised.

**Reply to review of Anonymous referee #2**

Dear reviewer,

Thank you for your positive and encouraging review of our manuscript. We agree with you that this study is a nice opportunity to extract more knowledge about different processes controlling runoff at different stages. Therefore, we really appreciate your detailed comments and suggestions, which can help us to improve the manuscript. We would like to respond to your comments below.

**Abstract**

*I think the abstract can be a bit longer. I miss this sentence from the Methods section to describe the analysis made in the experiment Three types of statistical analyses and comparisons of simulation results and observations were conducted: cumulative discharges, empirical extreme value distribution of the peak flows and flow duration curves for low flows. I would also recommend highlighting in the abstract more findings when it comes to understanding controls of these parts of the hydrograph, related to differences in model performance, as well as, the fact that some parts of the hydrograph is poorly understood and here we can thus identify present knowledge gaps.*

We are a bit hesitant to really increase the length of the abstract; however, we agree with you on the missing elements. Therefore, we have restructured the abstract in the revised version of the manuscript.

**Introduction**

*The first part is a bit too general and has many references to the literature, which are not necessary for the understanding of the paper and as a reader I miss guidance to how the citied papers actually contributed to the topics discussed. Often the references are lumped and support general statements that doesn't help the reader much in deeper understanding of the literature and previous work, see example below.*

*Page 2, row 12: reading "Hydrological studies at different scales and under different climates have shown a large variety of hypotheses on hydrological functioning (e.g., McDonnell, 2013; Zehe et al., 2013; Fenicia et al., 2013; Clark et al., 2016; Seibert et al., 2016)." This is an example of a very general statement with no guidance to what hypothesis the references refer to. Please, give examples and divide the lumped chain of references to explain to the author why these references are chosen what variety did they show in hydrological understanding?*

*Page 2 row 22: I suggest starting the Introduction here the paragraphs above don't contribute much but are common knowledge. This is where the study is motivated and from this point the text is much more interesting and straightforward. (Avoid reference dropping!)*

Replying to the three comments above, we agree that the first part of the introduction is rather long and general. However, we prefer to keep some of it to give guidance to readers with a smaller

modelling background. So, in the revised version of the manuscript we have reduced the length of the first part of the introduction and have made it more specific for the current study. In line with this, we have elaborated a bit further on the selected references.

*Page 2 row 31: This sentence is difficult to understand: Ceola et al. (2015) concluded that deriving the causes of performance differences between various model structures is not trivial, mainly due to the considerable differences in model structures which disturbs the identification of model features that increase model performance. Are you trying to explain the problem of equifinality here? (i.e. that many different parameter settings may result in similar model performance, due to compensating processes in the model description?) Please, rephrase!*

Thank you for pointing this out; however, equifinality is not what we are aiming at in this sentence. Rather, we wanted to point out that because model structures are rather complex, it is difficult to compare them in an intercomparison study and to derive conclusions on model performance based on the presence or absence of certain processes in the model schematisation. In our study, we tried to overcome this problem by presenting schematic pictures of the model structures in Figure 2 of the paper.

Therefore we have rephrased the sentence as follows: Ceola et al. (2015) pointed out that previous intercomparison studies have contributed little to deriving the causes of performance differences between various model structures. This could be attributed to the complexity and the large differences of model structures, and to the difficulty to link the presence of a model feature to a better or worse performance.

*Page 4 row 15: I guess the notation of 'local time' can be removed in this decadal context.*

Thank you for pointing this out, we have removed the notation of 'local time'.

**Method**

*The experiment is very straight forward and described with relevant level of details. I like the schematic pictures of model structures of Fig 2 and the descriptions in Table 2. Very useful for understanding!*

Thank you! Nice to hear that the figure is very useful.

**Results**

*I suggest including statistical metrics for model performance seen in Fig 3-9. Each graph can be assigned with a (few) metric(s) for the data it is showing, to support the analysis.*

See also our reply to the third comment of the first reviewer. We agree that adding metrics to the figures will indeed support the analysis. For the revised version of the manuscript we have added some metrics to the different plots that reflect the patterns we visually observed.

*Section 4.2 Modelling the highest peaks: please, elaborate on potential causes why some models are more successful than other to capture the peaks. Are there processes they did describe that others did not? Where they more carefully calibrated? (did they apply other considerations for*

*parameter choices?)*

It is a good point to have a more thorough look at whether the differences in capturing the highest peaks can be explained by differences in model structures. However as the results observed in the Lesse are not consistently observed in the other catchments (supplement, Section 4), it seems difficult to draw sound conclusions on this topic with regards to model structure components. Calibration certainly plays a role as shown in Fig 9 of the paper for the VHM model, where the same model performs better when another calibration objective is used. However, in Figure 4, all models were calibrated using the same objective functions, so we do not expect that calibration can really explain the differences between the models.

*Fig 3: it would be more interesting to see relative error of annual flow instead of absolute; at least for a reader who is not familiar with the catchment but with general model performances.*

We agree and have changed the figure.

*Section 4.5 Transition from low to high flows what can be learnt from these results of poor model performance? Interesting that a snow routine was not necessary, but could be compensated for how?*

Regarding the transition from low to high flows, most models show an overestimation of flow for most catchments. This indicates that the re-wetting of catchments works differently from what is currently assumed in the models. The variability in performance between catchments further indicates that the models are probably missing a process that is important during this stage of the hydrograph. We have elaborated on this in the revised manuscript.

Regarding the snow module: the influence of snowmelt on the discharge is small; however some snow cover does occur every winter. (generally snow accumulation is scattered in time and in the order of 20-50 mm SWE, and there are around 50 days with snow cover each year). Thus, by calibrating on NSE, many models were able to model the winter discharge right, even without a snow module. The effect of a snow module can better be shown and investigated when the same model is run with and without a snow module.

**Discussion**

*Section 5.1: With the title Findings about the Meuse basin I suggest you add some findings about which processes that seem to control flow of high peaks, low flow, etc. based on your results from model performance using different process descriptions/structures. Alternatively, you should change the title to Model performance in the Meuse basin but this will make the paper less appealing in my view. I would rather like to see the hydrological interpretation from the model results (i.e. change perspective to describe nature instead of models).*

We agree with you, Section 5.1 will benefit from a more detailed hydrological interpretation. In the revised manuscript we have linked the findings more to modelled runoff processes.

*Page 11, row 10: I suggest to rephrase We therefore hypothesize to We therefore suggestor rather: The results thus indicate. Even if results may raise new questions, I think you should take the*

*opportunity to make a statement from your study here.*

Thank you for the suggestion, we have rephrased the sentence.

*Section 5.2: Benefits of an intercomparison study: Please, specify more clearly what direct benefit you could draw from using a model ensemble in this catchment. It is a costly and time-consuming process, so what are the proofs from your results claiming that it is worthwhile?*

This study provided some clear conclusions for model elements (or runoff processes) that are important during drying conditions, although this stage of the hydrograph is often difficult to model. Following on this, the study reflects the importance of model structures (or choices) for different parts of the hydrograph. A model ensemble and the background on individual performance can help in operational forecast for, among others, uncertainty estimation.

*Page 11, row 28: I suggest acknowledging the importance of following a firm protocol (as you seem to have done here) in the collaboration, to ensure transparency for the result analysis and reproducibility of computational experiments. See for instance: Hutton, C., Wagener, T., Freer, J., Han, D., Duffy, C. and Arheimer, B., 2016. Most computational hydrology is not reproducible, so is it really science?. Water Resour. Res.. Accepted Author Manuscript. doi:10.1002/2016WR019285*

Thank you for sharing this reference, which we have included in the introduction.

*Page 12 row 6: Note that it is quite common that catchment models underestimate extremes even though they do reasonable good on mean values, also in a multibasin analysis. See for instance: Donnelly, C, Andersson, J.C.M. and Arheimer, B., 2016. Using flow signatures and catchment similarities to evaluate a multibasin model (E-HYPE) across Europe. Hydr. Sciences Journal 61(2):255-273, doi: 10.1080/02626667.2015.1027710*

Thank you for sharing this reference, which we have included in the paper.

**Conclusion**

*I would recommend highlighting your findings in understanding controls of the various parts of the hydrograph, related to differences in model performance. Was it only events during dry conditions you could refer to improved processes understanding? Check if there is more from your work to extract here! For Transition from low to high flows as well as extreme flows I think you should acknowledge that we see knowledge gaps in process understanding and identify weaknesses in the model descriptions.*

For transition and wet conditions, the results were not very consistent between the different catchments, revealing the presence of knowledge gaps in understanding model features that causes these differences. We have acknowledged these gaps in process understanding briefly in the conclusions.

**Reply to review of Rohini Kumar**

Dear Rohini Kumar,

Thank you for your positive evaluation of our manuscript. We value the comments and suggestions you have made and would like to respond to them below.

1. *Missing references (relevant to this study) - specifically of the distributed model intercomparison studies (phase 1 and 2), which have also analyzed model skill using event specific metrics for specifying inter-model differences. Reed, S., Koren, V., Smith, M., Zhang, Z., Moreda, F., Seo, D. J., & Participants, D. M. I. P. (2004). Overall distributed model intercomparison project results. Journal of Hydrology, 298(1), 27-60. Smith, M. B., Koren, V., Zhang, Z., Zhang, Y., Reed, S. M., Cui, Z., ... & Anderson, E. A. (2012). Results of the DMIP 2 Oklahoma experiments. Journal of Hydrology, 418, 17-48.*
   Thank you for pointing out these two related studies, we have consulted them and have included them in the revised version of the manuscript.

2. *I would suggest the authors to illustrate the modeling protocol in a step-wise manner or a flow chart to better follow the content.*
   We agree with you that a flow chart with the modelling protocol would make it easier to follow for the reader. In our reply to the first reviewer, we proposed a possible schematisation for this chart, which we included in the revised version of the manuscript.

3. *It was bit difficult for me to follow the results - switching from one catchment to another. The rational/reason behind presenting results in this way should be clear to the readers.*
   We understand that currently the selection of presented results might seem a bit arbitrary. However, in each section of the results, we selected a catchment that illustrated most clearly the message we wanted to convey. We have pointed out at the beginning of the results section why we have selected certain catchments. In addition, we have added a figure summarising the resuts for all models and catchments. The results for all catchments are also shown in the supplement to support the selection made for the manuscript itself.

4. *I would also suggest the authors to comprehensively present their results for different catchments in a tabular form.*
   We agree with you that in certain cases a table can be very informative to present results. However, we do not think that a table with NSE/NSElog values for all catchments will be very informative, as we also pointed out in our reply to the first reviewer. We have included a summarising figure instead of a table in the revised manuscript, as we think a figure is even more informative and often easier to interpret.

5. *Blind validation in space: How different are the selected catchments in terms of their (dynamic) hydrologic behaviors? - for example in terms of correlation metrics of daily/hourly stream flows*

Thank you for asking this question. In general, the hydrological behaviour of the catchments is similar, we have provide some more details about the daily/hourly stream flows in the revised version of the manuscript; in the text and in Table 1.

6. *Among all the selected models, the GR4H model appears to have the least (or almost no) uncertainty in model outputs. Please explain?*
   The band width in the 20 model realisations of GR4H is indeed very narrow. This is due to the calibration algorithm applied, which followed a stepwise calibration to an optimum after pre-filtering of the parameter space (see Table 2). This aspect of the results was discussed in section 5.3, however we have pointed this out in the result section as well.

7. *Section 4.2 Modelling the highest peaks could be revised as "Modelling the flood peaks"*
   Thank you for pointing this out, we have changed the title of this section.

[revised manuscript text omitted]

---

## Author Response (AR2)

Dear editor,

Thank you for your positive reply on our revisions and for handling our manuscript. Please find attached the final version.

Best wishes for the holiday season,

on behalf of all the authors,

Tanja de Boer-Euser